# Antigen Cross-Presentation by Murine Proximal Tubular Epithelial Cells Induces Cytotoxic and Inflammatory CD8^+^ T Cells

**DOI:** 10.3390/cells11091510

**Published:** 2022-04-30

**Authors:** Alexandra Linke, Hakan Cicek, Anne Müller, Catherine Meyer-Schwesinger, Simon Melderis, Thorsten Wiech, Claudia Wegscheid, Julius Ridder, Oliver M. Steinmetz, Linda Diehl, Gisa Tiegs, Katrin Neumann

**Affiliations:** 1Institute of Experimental Immunology and Hepatology, University Medical Center Hamburg-Eppendorf, 20246 Hamburg, Germany; alexandra_linke@web.de (A.L.); hakan.cicek@stud.uke.uni-hamburg.de (H.C.); anne.mueller1@uke.de (A.M.); c.wegscheid@web.de (C.W.); julius.ridder4u@gmail.com (J.R.); li.diehl@uke.de (L.D.); g.tiegs@uke.de (G.T.); 2Hamburg Center for Translational Immunology, University Medical Center Hamburg-Eppendorf, 20246 Hamburg, Germany; s.melderis@uke.de (S.M.); osteinmetz@uke.de (O.M.S.); 3Institute of Cellular and Integrative Physiology, University Medical Center Hamburg-Eppendorf, 20246 Hamburg, Germany; cmeyer-schwesinger@uke.de; 4III. Medical Clinic, University Medical Center Hamburg-Eppendorf, 20246 Hamburg, Germany; 5Institute of Pathology, University Medical Center Hamburg-Eppendorf, 20246 Hamburg, Germany; t.wiech@uke.de

**Keywords:** proximal tubular epithelial cells, antigen cross-presentation, cytotoxic CD8^+^ T cells, apoptosis, lupus nephritis

## Abstract

Immune-mediated glomerular diseases are characterized by infiltration of T cells, which accumulate in the periglomerular space and tubulointerstitium in close contact to proximal and distal tubuli. Recent studies described proximal tubular epithelial cells (PTECs) as renal non-professional antigen-presenting cells that stimulate CD4^+^ T-cell activation. Whether PTECs have the potential to induce activation of CD8^+^ T cells is less clear. In this study, we aimed to investigate the capacity of PTECs for antigen cross-presentation thereby modulating CD8^+^ T-cell responses. We showed that PTECs expressed proteins associated with cross-presentation, internalized soluble antigen via mannose receptor-mediated endocytosis, and generated antigenic peptides by proteasomal degradation. PTECs induced an antigen-dependent CD8^+^ T-cell activation in the presence of soluble antigen in vitro. PTEC-activated CD8^+^ T cells expressed granzyme B, and exerted a cytotoxic function by killing target cells. In murine lupus nephritis, CD8^+^ T cells localized in close contact to proximal tubuli. We determined enhanced apoptosis in tubular cells and particularly PTECs up-regulated expression of cleaved caspase-3. Interestingly, induction of apoptosis in the inflamed kidney was reduced in the absence of CD8^+^ T cells. Thus, PTECs have the capacity for antigen cross-presentation thereby inducing cytotoxic CD8^+^ T cells in vitro, which may contribute to the pathology of immune-mediated glomerulonephritis.

## 1. Introduction

Immune-mediated glomerular diseases are characterized by renal infiltration of T cells that localize in the tubulointerstitium in close contact to tubular epithelial cells. Interactions between lymphocytes and renal epithelial cell populations have been suggested to contribute to kidney disease pathology [1,2]. Proximal tubular epithelial cells (PTECs) have been described as a population of renal non-professional antigen-presenting cells (APCs), which induce antigen-specific CD4^+^ T-cell activation [3,4] and inflammatory cytokine expression [4]. PTECs are characterized by a low expression of major histocompatibility complex class (MHC)-II in homeostasis [3,5,6], which is upregulated in kidney diseases [5,7,8,9]. They present antigen via MHC-II [3], and express co-stimulatory molecules [4,10,11] thereby facilitating CD4^+^ T-cell activation. 

Cross-presentation is a process by which exogenous soluble antigens are presented through APCs on MHC-I to activate CD8^+^ T cells [12,13,14]. The cross-presentation pathway plays a key role in CD8^+^ T cell-mediated immunity to certain infections [15], tumors [16], and organ transplants [17]. It also contributes to autoimmune diseases where cross-presentation of self-antigens triggers autoimmunity [18]. Initially thought to be restricted to professional APCs such as dendritic cells (DCs) [19], cross-presentation is also efficiently carried out by liver sinusoidal endothelial cells (LSECs), a population of non-professional APCs in the liver [20,21]. Interestingly, antigen cross-presentation by LSECs induces CD8^+^ T-cell tolerance rather than immunity [22,23,24]. 

The major pathway for antigen cross-presentation is the endosome-to-cytosol pathway [25]. The mannose receptor (MR) has been implicated in the uptake of soluble antigen and its endosomal localization [26,27]. Internalized antigens are transported from endosomes into the cytosol for proteasomal degradation. Antigen translocation into the cytosol involves the valosin-containing protein (VCP), an ATPase that provides energy for antigen transport across endosomal membranes [28]. The trimeric translocon Sec61 has been suggested as an endosomal transmembrane pore complex crucial for antigen translocation [29,30]. The immunoproteasome and its specific subunit large multifunctional peptidase (LMP7) have been implicated in the processing of antigenic peptides for presentation on MHC-I [31,32]. Proteasome-derived peptides are transported by the transporter associated with antigen processing (TAP1) and TAP2 from the cytosol back into endosomes [33], where they are loaded onto MHC-I. Therefore, peptides are trimmed into suited sizes via the endosomal leucyl-cystinyl aminopeptidase (LNPEP) [34]. To avoid pH-dependent activation of proteases leading to impaired antigen cross-presentation, alkalization of endosomes is mediated by the NADPH oxidase (NOX)2 [35,36]. Subsequently, MHC-I/peptide complexes are transported to the cell surface for presentation to CD8^+^ T cells. 

Whether PTECs have the capacity to activate CD8^+^ T cells via antigen cross-presentation and what phenotype PTECs induce in CD8^+^ T cells is not known so far. Identifying the nature of PTEC-induced CD8^+^ T-cell responses may help to understand the impact of renal epithelial cell-mediated modulation of infiltrating T cells on the pathogenesis of immune-mediated glomerulonephritis (GN). In this study, we describe the potential of PTECs for antigen cross-presentation in vitro, identify pathways of soluble antigen internalization and processing, determine the phenotype of PTEC-activated CD8^+^ T cells in co-culture, and analyze the capacity of PTEC-activated CD8^+^ T cells to induce apoptotic cell death. We further assess the impact of CD8^+^ T cells on the induction of apoptosis in renal tissue in murine lupus nephritis.

## 2. Material and Methods

### 2.1. Animals 

Mice of the following types were bred in the animal facility of the University Medical Center Hamburg-Eppendorf (UKE; Hamburg, Germany): C57BL/6 wild-type (WT), *Cd8a*^-/-/-^, and C57BL/6-Tg(TcraTcrb)1100Mjb/Crl (OT-I). A selection of OT-I mice were kindly provided by Dr. Dorothee Schwinge and Prof. Hans-Willi Mittrücker (all UKE, Hamburg, Germany). The MRL/MpJ (MRL) and the MRL/MpJ-*Fas*^lpr^/J (MRL-*lpr*) mice were purchased at the Jackson Laboratory (Bar Harbor, ME). Mice were bred according to the Federation of European Laboratory Animal Science Association guidelines, and they were maintained under specific-pathogen-free conditions. All mouse experiments were approved by the Behörde für Justiz und Verbraucherschutz (Hamburg, Germany; approval codes: N57/19, ORG 960, and ORG 1032), and they were carried out according to the current existing guidelines on mouse experimentation. All efforts were made to minimize suffering.

### 2.2. Animal Treatment

The WT mice and the *Cd8a*^-/-^ mice received a single intraperitoneal injection of 500 µL pristane oil (2,6,10,14-tetramethylpentadecane; Sigma-Aldrich, St. Louis, MO, USA) to induce lupus nephritis, and were analyzed 9 months later. 

### 2.3. Urine Analysis

One day before the final analysis, mice were housed in metabolic cages for urine collection. Albumin levels were determined by ELISA (Mice-Albumin Kit; Bethyl, Montgomery, TX). Creatinine levels were measured with the COBAS INTEGRA Creatinine Jaffé Gen.2 Kit (Roche Diagnostics, Indianapolis, IN, USA). The albumin-to-creatinine ratio was calculated to assess the severity of proteinuria. 

### 2.4. Histological Analyses

To identify CD8^+^ CD3^+^ T cells in renal tissue, CD8 and CD3 staining was performed in serial sections. Therefore, 2 µm paraffin-embedded kidney sections were stained with anti-CD3 (polyclonal, Agilent Technologies, Santa Clara, CA, USA) or with anti-CD8 (D4W2Z; Cell Signaling Technology, Danvers, MA, USA) antibodies. Prior to CD8 staining, heat-induced antigen retrieval was achieved by using a citrate buffer (pH 6). A horseradish peroxidase (HRP)-conjugated anti-rabbit antibody (ThermoFisher Scientific, Waltham, MA, USA) was used as a secondary antibody. The DAB+ Substrate Chromogen System (Agilent Technologies) was used for antigen detection. Prior to CD3 staining, heat-induced antigen retrieval was done by using the Dako Target Retrieval Solution (Agilent Technologies). For antigen detection, New Fuchsin (Merck, Darmstadt, Germany) and the ZytoChem-Plus AP Polymer-Kit (Zytomed Systems, Berlin, Germany) were used, containing an alkaline phosphatase-conjugated anti-rabbit secondary antibody. To assess the number and the localization of CD8^+^ T cells, kidney sections were scanned using the Zeiss Axioscan 7 (Carl Zeiss, Jena, Germany), and they were analyzed by ZEN lite software (Carl Zeiss). Five corresponding high power fields (hpf) were randomly defined in the cortices of each serial kidney section. Overlapping CD3 and CD8 staining marked CD8^+^ T cells. 

For TUNEL staining, 3 µm paraffin-embedded kidney sections were prepared. The TUNEL assay was performed with the TACS Blue Label In Situ Apoptosis Detection Kit (R&D Systems, Minneapolis, MN) according to the manufacturer’s manual. To stain cleaved caspase-3, 2 µm paraffin-embedded kidney sections were incubated with an anti-cleaved caspase-3 antibody (ThermoFisher Scientific). Antigen retrieval was performed in Dako Target Retrieval Solution (Agilent Technologies). After incubation with goat anti-rabbit-HRP secondary antibody (ThermoFisher Scientific), kidney sections were stained with DAB+ substrate (Agilent Technologies). The evaluation was carried out by counting positive events in three random areas (each 125 µm^2^) located in renal cortex, from which the mean value was calculated.

Periodic acid-Schiff (PAS) staining was done with 1.5 µm paraffin-embedded kidney sections to assess disease-related pathological changes. The severity of lupus nephritis in MRL-*lpr* and MRL mice was evaluated according to the modified National Institute of Health activity and chronicity indices by a renal pathologist in a blinded manner. The renal activity score comprised glomerular proliferation, leucocyte exudation, karyorrhexis/fibrinoid necrosis (×2), cellular crescents (×2), hyaline deposits and interstitial inflammation [37]. Glomerular abnormalities in pristane-treated mice were determined in a minimum of 50 glomeruli per mouse in a blinded manner. These included glomerular hypercellularity, crescent formation, fibrinoid necrosis, segmental proliferation, hyalinosis and capillary wall thickening [38]. Glomeruli were scored as severely abnormal if they showed intra and/or extra-capillary proliferation or crescents. Glomeruli were scored as abnormal if they showed mesangial proliferation and/or increased mesangial deposition of PAS-positive material or if they fulfilled criteria for severely abnormal glomeruli.

### 2.5. Isolation and Culture of PTECs

Kidneys were harvested from WT mice. Renal medulla and adrenal structures were removed and the remaining cortices were finely minced. Renal cortex tissue was digested in DMEM/F-12/GlutaMAX^TM^ medium (ThermoFisher Scientific), containing 0.25% BSA (Serva Electrophoresis GmbH, Heidelberg, Germany) and 0.01% collagenase from *clostridium histolyticum* (Sigma Aldrich, St. Louis, MO, USA) at 37 °C for 17 min. Thereafter, renal tissue was passed through a 250 µm sieve. PTECs were separated from other cortical cells by Percoll density gradient centrifugation, using a solution that contained 45% Percoll (GE Healthcare Life Sciences, Chicago, IL, USA) and 55% 2× PBS-glucose as described previously [4,39]. After gradient centrifugation, PTECs were harvested from the lowest interphase and cultured in DMEM/F12 media supplemented with 1% FCS, 1× I/T/S (mixture of insulin, transferrin, sodium selenite), 50 nM hydrocortisone, 5 nM T_3_, 5 nM epidermal growth factor (all Sigma Aldrich), and 1% penicillin/streptomycin (ThermoFisher Scientific) for 5 days to form a monolayer. PTECs were incubated with Alexa Fluor 647-conjugated ovalbumin (OVA; 10 µg/mL; ThermoFisher Scientific) at 37 °C or 4 °C. PTECs were pre-treated with chlorpromazine (30 µM, [40]), mannan (3 mg/mL [26]) or D-galactose (0.125 M [41]; all Sigma-Aldrich) at 37 °C for 1 h before incubation with Alexa Fluor 647-labelled OVA for another hour.

### 2.6. Isolation and Culture of LSECs 

Livers of WT mice were perfused with 0.05% GBSS/collagenase (from *clostridium histolyticum*; Sigma Aldrich) solution, removed, and digested in GBSS/collagenase at 37 °C for 20 min. After filtration through a steel mesh, liver cells were washed twice with GBSS. Non-parenchymal cells were removed by density gradient centrifugation, using a 30% Nycodenz solution (Progen Biotechnik, Heidelberg, Germany). By magnetic-activated cell sorting (MACS), CD146^+^ LSECs were isolated using CD146 MicroBeads (Miltenyi Biotec, Bergisch Gladbach, Germany). Then, LSECs were seeded onto collagen-coated 24-well plates in DMEM (high glucose) supplemented with 8% FCS (Sigma Aldrich), 2% L-glutamine (Life Technologies, Carlsbad, CA, USA), and 1% penicillin/streptomycin (ThermoFisher Scientific) for 2 days to form a monolayer.

### 2.7. Generation of Bone Marrow-Derived DCs (BMDCs)

Bone marrow cells were isolated from femur and tibia of WT mice. After erythrocyte lysis through NH_4_Cl, 1 × 10^6^ cells were seeded in a petri dish in the presence of GM-CSF (10 ng/mL; PeproTech, Rocky Hill, NJ, USA) for 7–10 days. Thereafter, CD11c^+^ BMDCs were isolated by MACS using CD11c MicroBeads (Miltenyi Biotec).

### 2.8. Isolation of CD8^+^ T Cells and Splenic DCs 

From the spleens and the lymph nodes of OT-I mice, OVA-specific CD8^+^ T cells were isolated; DCs were isolated from the spleens of WT mice. Tissue was passed through 70 µm nylon meshes prior to erythrocyte lysis using NH_4_Cl. Thereafter, cells were incubated with anti-CD16/32 antibody solution. Using the CD8^+^ T cell Isolation Kit (Miltenyi Biotec) according to the manufacturer’s instructions, CD8^+^ T cells were enriched by MACS. Subsequently, CD8^+^ CD25^−^ T cells were purely isolated by fluorescence-activated cell sorting (FACS). Using CD11c Microbeads, CD11c^+^ DCs were isolated by MACS. 

### 2.9. Isolation of Renal Cells

Murine kidneys were harvested, finely minced, and digested for 40 min at 37 °C with 0.4 mg/mL collagenase D and 0.01 mg/mL DNase I (both Roche). Single cell suspensions were achieved by using the gentleMACS Dissociator (Miltenyi Biotec). Cell suspensions were subjected to a density gradient centrifugation, using a 37% Percoll solution (GE Healthcare Life Sciences) to enrich renal leukocytes. Erythrocytes were lysed with NH_4_Cl. 

### 2.10. Co-Culture Experiments

For 2.5 or 5 days, OVA-specific CD8^+^ CD25^‒^ T cells were co-cultured with PTECs, splenic DCs, or LSECs in the presence or the absence of peptide-free OVA protein (500 µg/mL). Peptide-free OVA protein was generated using Amicon Ultra-15 centrifugal filter devices (Merck), according to the manufacturer’s instruction. For immunoproteasome inhibition, PTECs were pre-incubated with ONX 0914 (0.1 µM; Cayman Chemical, Ann Arbor, MI) for four hours. All PTECs were thoroughly washed before co-culture with CD8^+^ T cells in the presence and the absence of OVA for 1.5 days. 

### 2.11. Flow Cytometry

The CD8^+^ T cells from co-cultures and renal leukocytes were re-stimulated with phorbol myristate acetate (10 ng/mL) and ionomycin (250 ng/mL) for 4 h with the addition of brefeldin A (1 µg/mL; all Sigma Aldrich) and monensin (BioLegend, San Diego, CA) after 30 min. The anti-CD107a antibody (1D4B; FITC; BioLegend) was added to the restimulation medium. Cells were incubated with anti-CD16/32 antibody solution (93; BioLegend, San Diego, CA, USA) to prevent unspecific antibody binding. LIVE/DEAD Fixable Staining Kits (ThermoFisher Scientific) were used to exclude dead cells. Cells were surface-stained with fluorochrome-labeled antibodies specific to T-cell receptor (TCR)β (H57-597; FITC), CD8a (53-6.7; V500), CD25 (PC61; BV421), CD44 (IM7; BV785), and PD-1 (HA2-7B1; APC; all Biolegend). After fixation and permeabilization (Foxp3 Transcription Factor Staining Buffer Set; ThermoFisher Scientific), CD8^+^ T cells were stained with antibodies specific to Ki-67 (REA183; FITC), granzyme B (GB11; Pacific Blue), interferon (IFN)γ (XMG1.2; PE- Texas Red), or interleukin (IL)-17A (TC11-18H10.1; PerCP; all Biolegend). To determine cytokine levels in culture supernatants via flow cytometry, the LEGENDplex MU Th Cytokine Panel (12-plex, Biolegend) was used. 

### 2.12. Cytotoxicity Assay

As P815 mastocytoma cells are well described target cells for cytotoxic T cells [42], they were therefore used in the cytotoxicity assay. In a target/effector-cell ratio of 1:6 for 4 h, 1 × 10^5^ P815 cells were cultured with PTEC-activated CD8^+^ T cells or the respective controls. Cells were stained for CD8 (53-6.7; BV785; BioLegend) to distinguish CD8^-^ P815 cells from CD8^+^ T cells. For dead cell discrimination, cells were stained with 7-amino-actinomycin D (7-AAD; PerCP; Biolegend), which binds to the DNA of damaged and dead cells, and it is excluded by intact cells. The staining of CD8 and 7-AAD was done 15 min before data acquisition without cell washing. 

### 2.13. Quantitative Real-Time RT- PCR Analysis

Total RNA was isolated from cultured PTECs, BMDCs, and LSECs using Trizol reagent (ThermoFisher Scientific). The RNA was subjected to DNase (ThermoFisher Scientific) digestion to remove contaminating genomic DNA. The RNA was transcribed into cDNA using the Verso cDNA Synthesis Kit (Life Technologies) on a MyCycler thermal cycler (BioRad, München, Germany). A quantitative RT-PCR was performed using the Absolute qPCR SYBR Green Mix (ThermoFisher Scientific). The relative mRNA levels were calculated using the ∆∆CT method after normalization to the reference gene β-actin. Exon-spanning primers were obtained from Metabion (Martinsried, Germany). Primer sequences are listed in Appendix A. 

### 2.14. Western Blot Analysis

Cultured PTECs, BMDCs, or LSECs were incubated in a lysis solution containing 1% Triton X-100 (Sigma Aldrich). To prevent proteolytic degradation, incubation was done in the presence of the protease and the phosphatase inhibitors EDTA (5 mM), PMSF (1 mM), aprotinin, sodium pyrophosphate (10 mM), beta-glycerophosphate (10 mM), sodium orthovanadate (1 mM), sodium pervanadate (0.1 mM), and sodium fluoride (10 mM). Protein concentrations were determined with the Bradford assay. Equal protein levels were used and applied to precast NuPAGE Bis-Tris Gels with a polyacrylamide concentration of 4–12% (ThermoFisher Scientific) for protein separation. Proteins were transferred to a nitrocellulose membrane by tank blotting. Membranes were incubated with anti-TAP1, anti-SEC61A1, anti-NOX2, anti-LNPEP, and anti-VCP (all polyclonal; ThermoFisher Scientific), or with anti-MMR/CD206 (polyclonal, R&D Systems) antibodies. The secondary anti-rabbit or anti-goat antibodies were conjugated to horseradish peroxidase, and binding was visualized by chemiluminescence (ThermoFisher Scientific). To detect the reference protein GAPDH, blots were stripped and incubated with anti-GAPDH (6C5cc; HyTest, Turku, Finland) antibody. To perform a densitometric analysis, blots were analyzed with the Image Lab software (Bio-Rad, Feldkirchen, Germany). Signal intensities of the analyzed proteins were normalized to the reference protein GAPDH. 

### 2.15. Proteasomal Subunit Activity Measurement

Shock frozen pellets of PTECs pre-treated with ONX 0914 (0.1 µM) for 4 h or of rat hybridoma assay control cells were re-suspended in 12 µL TSDG buffer (10 mM Tris-HCl pH 7.5, 10 mM NaCl, 25 mM KCl, 1 mM MgCl_2_, 0.1 mM EDTA, 1 mM DTT, 2 mM ATP, 10% glycerol), and lysed by 7 freeze-thaw cycles using a mixture of dry ice and ethanol and a water bath. Lysates were centrifuged at 16,000 *g* for 10 min at 4 °C, supernatant volumes were measured and exact volumes were transferred to fresh tubes and filled up to 19.6 µL with TSDG buffer. Each sample was incubated in a reaction volume of 20 µL after the addition of 0.5 µM Cy5-tagged pan-proteasomal activity-based probes (ABP) for one hour at 37 °C. Rat hybridoma assay controls were prepared as follows: 5 µg total protein extract was incubated (a) without proteasomal inhibitor and ABP, (b) 0.5 µM ABP, (c) 2 µM epoxomicin (Enzo, New York, NY1 h, 37 °C), followed by 0.5 µM ABP, (d) 2 µM ONX 0914 (1 h, 37 °C), followed by 0.5 µM ABP or (e) 0.1 µM ONX 0914 (1 h, 37 °C), and followed by 0.5 µM ABP. PTECs were incubated with 0.5 µM ABP. Thereafter, samples were mixed with a reducing sample buffer (50 mM Tris-HCl pH 6.8, 2% SDS, 100 mM DTT, 10% glycerol, 0.05% bromphenol blue) and boiled at 70 °C for 10 min. Samples were separated by SDS PAGE (12.5% self-cast gel from a 29:1 acrylamide/bis solution; Serva, Heidelberg, Germany) and imaged using a Vilber Fusion FX07 system (light capsule C640 and Filter F710) for measurement of proteasomal subunit activities. After that, samples were blotted on a PVDF membrane (Merck), and they were incubated with antibodies for the quantification of proteolytic proteasomal subunit protein abundance. The following antibodies were used: rabbit anti-β1c (PSMB6), rabbit anti-β5c (PSMB5, both Invitrogen), and rabbit anti-β5i (Lmp7; self-made, Elke Krüger, Biochemistry Greifswald, Germany). To assess total protein abundance, a mouse anti-β-actin (Sigma Aldrich) antibody was used. Binding of the ABP to active proteasomal subunits induced a MW shift of a few kDa in comparison to the inactive proteasomal subunit to which no ABP was bound.

### 2.16. Sequencing Data

The RNA-seq data set of renal epithelial cells and hepatic endothelial cells isolated from WT mice were available from the NCBI Gene Expression Omnibus (GEO) repository (GSE134663), and they were originally published by Krausgruber et al. [43]. We defined a gene set of 10 genes, all associated with antigen cross-presentation. The means of Log_2_cpm (counts per million; n = 3), which represents the respective gene expression level within the gene set, were shown in heat maps. 

### 2.17. Statistical Analysis

Data were analyzed using the GraphPad Prism software (GraphPad software, San Diego, CA, USA). Statistical comparison was carried out using a Mann–Whitney U test or a one-way ANOVA with post analysis by Tukey–Kramer test. Data were presented as medians. A *p* value of less than 0.05 was considered statistically significant with the following ranges * *p*< 0.05, ** *p*< 0.01, *** *p*< 0.001, and **** *p*< 0.0001.

## 3. Results

### 3.1. PTECs Express Proteins Crucial for Antigen Cross-Presentation

There is an increasing body of evidence that non-hematopoietic, tissue-resident cell populations, such as endothelial and epithelial cells, are regulators of organ-specific immune responses [43]. To address a potential contribution of renal epithelial cells to the modulation of CD8^+^ T-cell responses, we created a gene expression analysis of epithelial cell adhesion molecule (EpCAM)^+^ epithelial cells isolated from the kidneys of C57BL/6 mice by using a previously published RNA-seq dataset [43]. For our analysis, we used the Log_2_cpm, which represent the gene expression levels, from a gene set of 10 genes associated with antigen cross-presentation. We determined strong gene expression of the Sec61 translocon subunits (*Sec61a1*, *Sec61b, Sec61g*), the ATPase VCP (*Vcp*), and the aminopeptidase LNPEP (*Lnpep*) as well as substantial expression of the immunoproteasome subunit LMP7 (proteasome beta type (*Psmb*)*8*) and MR (*Mrc1*), whereas the TAP transporters (*Tap1, Tap2*) and the NADPH oxidase NOX2 (*Cybb*) were less expressed (Figure 1A). The same analysis was done with CD31^+^ endothelial cells from the liver, from which it has been shown that particularly LSECs have a high capacity for antigen cross-presentation [20,21,22,23,24]. Here, we found stronger expression of *Psmb8*, *Mrc1*, and *Tap1*, while *Sec61a1* and *Sec61b* were less expressed (Figure 1B). Thus, both renal epithelial cells and hepatic endothelial cells express genes associated with antigen cross-presentation.

Within the different epithelial cell populations of the kidney, PTECs have been described as renal non-professional APCs that stimulate inflammatory CD4^+^ T-cell activation [3,4]. Therefore, we asked whether PTECs are also able to activate CD8^+^ T cells by antigen cross-presentation. We assessed expression of genes associated with cross-presentation in cultured PTECs and compared their expression profile with BMDCs, a population of professional APCs, and LSECs—a population of hepatic non-professional APCs—both capable of cross-presenting exogenous antigen to CD8^+^ T cells [44,45]. Similar to the gene expression data of total renal epithelial cells, we found that PTECs expressed all genes that were included in our analysis. However, PTECs showed much less expression of *Mrc1* and *Tap1* in comparison to BMDCs and LSECs, whereas the gene expression levels of *Sec61b*, *Sec61g*, *Tap2*, and *Lnpep* were similar. PTECs and BMDCs showed a comparable expression of *Sec61a1* and *Vcp*, while the expression of both was increased in LSECs (Figure 1C). We then analyzed the expression of proteins involved in antigen cross-presentation in PTECs, comparing it to BMDCs and LSECs. We determined the expression of MR, TAP1, VCP, SEC61A1 (representative for the trimeric translocon), LNPEP, and NOX2 in PTECs. They showed a reduced MR expression compared to BMDCs and LSECs, while expression of TAP1 and NOX2 was elevated. The expression of SEC61A1 was enhanced in PTECs and LSECs compared to BMDCs, whereas VCP and LNPEP were expressed at the same level in all three cell populations (Figure 1D, Appendix A). Thus, PTECs express genes and proteins associated with antigen cross-presentation.

### 3.2. PTECs Internalize Soluble Antigen via MR-Dependent Endocytosis

A prerequisite for antigen cross-presentation is internalization of exogenous antigen. To test the ability of PTECs for soluble antigen uptake, we incubated PTECs with the fluorochrome-labelled model antigen OVA at 37 °C or 4 °C at different periods of time. We determined an increased frequency of OVA^+^ PTECs at 37 °C over time, which was strongly reduced after incubation at 4 °C (Figure 2A). The MR is an endocytic receptor, which has been implicated in soluble antigen uptake and cross-presentation [26,46]. To assess whether PTECs internalize OVA via the MR, they were pre-incubated with mannan, a polysaccharide that binds with a high affinity to the MR thereby competitively inhibiting MR-mediated endocytosis [41,47]. After the mannan treatment, the frequency of OVA^+^ PTECs decreased compared to PTECs that were not pre-incubated with mannan. In contrast, pre-treatment of PTECs with D-galactose—a polysaccharide that does not bind to the MR [41]—did not affect uptake of OVA. Since internalization of the MR involves clathrin-coated vesicles, we analyzed the effect of the clathrin-specific inhibitor chlorpromazine (CPZ) [48] on antigen uptake through PTECs. We showed that the pre-incubation of PTECs with CPZ reduced their ability to take up OVA (Figure 2B). Thus, PTECs internalize soluble antigen via MR-mediated endocytosis. 

### 3.3. PTECs Induce a Cytotoxic and Inflammatory Phenotype in CD8^+^ T cells

We next analyzed the capacity of PTECs to induce an antigen-dependent activation of CD8^+^ T cells via antigen cross-presentation. Therefore, we isolated OVA-specific CD8^+^ CD25^‒^ T cells from the spleen and the lymph nodes of OT-I mice (Appendix A), and we cultured them with PTECs in the presence or the absence of the antigen OVA for 2.5 days. As controls, CD8^+^ CD25^‒^ T cells were cultured alone in the presence or the absence of OVA protein. To assess CD8^+^ T-cell activation and proliferation, we stained for the activation markers CD25, CD44, and PD-1 as well as for the proliferation marker Ki-67. Without PTECs, CD8^+^ T cells did not show any activation or proliferation. Moreover, PTECs did not induce substantial T-cell activation in absence of OVA. In contrast, we determined elevated frequencies of CD8^+^ T cells expressing CD25, CD44, PD-1, and Ki-67 in the presence of PTECs and OVA (Figure 3A). We further analyzed the phenotype of PTEC-activated CD8^+^ T cells, showing expression of the cytotoxic molecule granzyme B (GzmB). Since cytotoxic CD8^+^ T cells store GzmB in granules, they must degranulate for its release to induce apoptosis in target cells. Expressed on membranes of such granules, CD107a has been described as a degranulation marker, since it can only be detected on the cell surface after the fusion of granules with the cell membrane during degranulation [49,50]. We detected an elevated frequency of CD8^+^ T cells expressing CD107a on the cell surface after activation through PTECs (Figure 3A), demonstrating their capacity for degranulation. 

By determining cytokine levels in culture supernatants, we found that CD8^+^ T cells cultured alone did not express cytokines (data not shown). In contrast, we detected IL-2 and the inflammatory cytokines IFNγ, tumor necrosis factor (TNF)α, IL-17A, and IL-6 in co-culture supernatants of PTEC-activated CD8^+^ T cells (Figure 3B), whereas anti-inflammatory cytokines such as IL-10 or type 2 cytokines such as IL-5 and IL-13 were not expressed (data not shown). We also performed a phenotype analysis of CD8^+^ T cells after 5 days of culture. Again, CD8^+^ T cells were not activated in the absence of PTECs. We showed that CD8^+^ T cells were still activated, proliferated, and expressed CD107a and GzmB in the presence of PTECs and OVA (Appendix A). We further determined the same cytokine expression profile compared to CD8^+^ T cells that were cultured for 2.5 days (Appendix A). Thus, PTECs induce an antigen-specific activation and proliferation of CD8^+^ T cells, which express cytotoxic molecules and inflammatory cytokines. 

### 3.4. PTECs Induce a Similar Phenotype in CD8^+^ T Cells as Professional APCs

To assess the potential of PTECs for CD8^+^ T-cell activation, we compared the phenotype of PTEC-activated CD8^+^ T cells with CD8^+^ T cells that were stimulated by professional DCs. Therefore, OVA-specific CD8^+^ CD25^‒^ T cells were cultured with splenic DCs in the presence or the absence of OVA for 2.5 days. We performed viSNE analysis [51] to visualize the phenotype of CD8^+^ T cells, which were either activated by PTECs or DCs. Clustering of CD8^+^ T cells stained for activation and proliferation markers is depicted. We detected a similar clustering of PTEC- and DC-activated CD8^+^ T cells with one cluster characterized by a strong co-expression of CD44 and Ki-67, while the other cluster did not express Ki-67. The expression of CD25 co-localized with CD44 was much lower than CD44 expression; PD-1 expression was mainly determined in the Ki-67^‒^ cluster (Figure 4A). In another viSNE analysis, we showed a comparable clustering of CD8^+^ T cells stained for markers associated with cytotoxicity. In both PTEC- and DC-activated CD8^+^ T cells, we found CD107a^+^ GzmB^low^ and CD107a^low^ GzmB^+^ clusters (Figure 4A), demonstrating the presence of degranulated and not degranulated CD8^+^ T cells, which differ in their amount of intracellularly stored GzmB. 

We showed a decreased frequency of PTEC-activated CD8^+^ T cells expressing CD25 and Ki-67 compared to CD8^+^ T cells activated by DCs. Both CD8^+^ T-cell populations showed the same expression of CD44, PD-1 and GzmB, whereas the expression of CD107a was elevated in PTEC-activated CD8^+^ T cells (Figure 4B). Additionally, DCs did not induce an expression of Th2 cytokines or IL-10 in CD8^+^ T cells (data not shown). However, DCs induced a stronger expression of IFNγ, while IL-17A and TNFα levels were comparable. Interestingly, we detected high levels of IL-6 in PTEC co-culture supernatants, whereas in DC co-cultures, IL-6 was not detectable (Figure 4C). After 5 days of co-culture, PTEC-activated CD8^+^ T cells still proliferated less and showed a reduced expression of PD-1, CD107a, and IFNγ than CD8^+^ T cells activated by DCs, while expression of GzmB was comparable (Appendix A). In contrast, IL-2 and TNFα level were increased in PTEC/CD8 co-culture supernatants (Appendix A). Thus, PTEC-stimulated CD8^+^ T cells are less activated than DC-stimulated CD8^+^ T cells, but they show a similar cytotoxic and inflammatory phenotype. 

### 3.5. Different Outcome of CD8^+^ T-Cell Activation Induced by Non-Professional APCs from Kidney and Liver

To assess the organ-specific characteristics of non-professional APC-mediated CD8^+^ T-cell activation, we did a comparative phenotype analysis of CD8^+^ T cells that were either activated by PTECs or LSECs. Therefore, OVA-specific CD8^+^ CD25^‒^ T cells were cultured with LSECs in the presence or the absence of OVA for 2.5 days. Compared to PTECs, LSECs induced the same level of activation, proliferation, and expression of cytotoxic molecules and inflammatory cytokines in CD8^+^ T cells. In contrast, IL-2 levels were decreased in LSEC co-cultures (Appendix A). Interestingly, we found a striking difference in the phenotype of activated CD8^+^ T cells after 5 days of co-culture. While CD8^+^ T cells activated by PTECs still proliferated and expressed cytotoxic molecules and inflammatory cytokines, LSEC-stimulated CD8^+^ T cells were still activated but no longer proliferated or expressed GzmB and CD107a (Figure 5A). They further showed a strongly reduced expression of IFNγ in comparison to PTEC-activated CD8^+^ T cells. Moreover, we did not detect IL-2 in LSEC co-culture supernatants (Figure 5B), most likely through consumption of the little IL-2 that was produced in the beginning by CD8^+^ T cells. Thus, in contrast to LSECs, PTECs do not have the ability to inhibit initial inflammatory CD8^+^ T-cell activation. 

LSECs have been shown to induce inhibition of effector CD8^+^ T cells by the programmed cell death (PD)-1/PD-L1 pathway [52,53]. While expression of the co-inhibitory receptor PD-1 was comparable in PTEC- and LSEC-activated CD8^+^ T cells after short culture time (Appendix A), we determined an elevated expression of PD-1 by LSEC-stimulated CD8^+^ T cells after 5 days of co-culture (Figure 5A). Moreover, the frequency of LSECs expressing the co-inhibitory ligand PD-L1 increased compared to PTECs, whereas expression of the co-stimulatory molecules CD80 and CD86 was mainly observed in PTECs (Figure 5C). Thus, PTECs and LSECs differ in their expression of co-inhibitory and co-stimulatory molecules.

### 3.6. PTEC-Induced CD8^+^ T-Cell Activation Depends on Proteasome Activity

The immunoproteasome plays a crucial role in antigen processing and subsequent T-cell activation; and, thus has become a target in autoimmune diseases [54]. To assess a potential contribution of the immunoproteasome in antigen cross-presentation by PTECs, we analyzed gene and protein expression of LMP7, a catalytic subunit selectively present in the immunoproteasome. We showed gene expression of *Psmb8* and protein expression of LMP7 by PTECs (Figure 6A,B). In comparison to BMDCs and LSECs, PTECs expressed much less LMP7 (Figure 6B). The pre-treatment of PTECs with the LMP7 inhibitor ONX 0914 [54] impaired PTEC-induced CD8^+^ T-cell activation. We determined a reduced frequency of activated and proliferating CD8^+^ T cells, and the expression of GzmB was also strongly decreased. The analysis was done after 1.5 days of co-culture, and at this time point the degranulation capacity of PTEC-activated CD8^+^ T cells was still minor as demonstrated by the low frequency of CD107a^+^ CD8^+^ T cells, which was further reduced after pre-treatment of PTECs with ONX 0914 (Figure 6C). Moreover, cytokine levels in co-culture supernatants were strongly diminished (Figure 6D), further demonstrating the impaired capacity of PTECs to induce inflammatory CD8^+^ T-cell activation after pre-incubation with ONX 0914. 

To address the specificity of ONX 0914-induced proteasome inhibition in PTECs, proteolytic activities and the protein expression of subunits present in the immunoproteasome or the constitutive proteasome were analyzed. In western blot (WB) analysis, we showed a higher abundance of the immunoproteasome-specific subunit β5i (LMP7) in ONX 0914-treated PTECs compared to untreated PTECs. However, protein abundance of β5c and β1c was also enhanced, which are two subunits of the constitutive proteasome (Figure 6E, Appendix A). We further determined a complete binding of both β5i and β5c by ONX 0914 (Figure 6E). We used ABPs that react with proteasomes in relation to their catalytic activity to assess proteolytic activity of the different proteasome subunits. The treatment of PTECs with ONX 0914 resulted in the strongly reduced activity of β5i and β5c, whereas β1c and β2c activity was not substantially altered (Figure 6E, Appendix A). Thus, by binding to β5i and β5c, ONX 0914 not only inhibited the proteolytic activity of an immunoproteasome subunit but also of a subunit present in the constitutive proteasome in PTECs.

### 3.7. PTEC-Activated CD8^+^ T Cells Exert Cytotoxic Function

PTEC-activated CD8^+^ T cells express the cytotoxic molecule GzmB and the degranulation marker CD107a, indicating that PTECs induce cytotoxicity in CD8^+^ T cells. To analyze the cytotoxic function of PTEC-activated CD8^+^ T cells, we performed a cytotoxicity assay with allogenic P815 target cells [42]. In this assay, cytotoxic CD8^+^ T cells become activated through interaction of their TCR with allogenic MHC molecules on P815 cells, resulting in induction of cell death in the target cells. Therefore, PTEC-activated CD8^+^ T cells were cultured with P815 cells for four hours. Subsequently, target cell killing was assessed by 7-AAD staining. As controls, P815 cells were incubated with CD8^+^ T cells pre-cultured with or without PTECs in the presence or the absence of OVA. Thereafter, the frequency of 7-AAD^+^ P815 cells was analyzed. Compared to P815 cells cultured alone, we determined no increased P815 cell death in the presence of CD8^+^ T cells pre-cultured without PTECs. The frequency of 7-AAD^+^ P815 cells increased if CD8^+^ T cells from PTECs pre-cultures without OVA were used. Interestingly, we determined a strongly elevated frequency of the 7-AAD^+^ P815 cell in the presence of CD8^+^ T cells that were pre-cultured with PTECs and OVA (Figure 7), demonstrating the cytotoxic function of PTEC-activated CD8^+^ T cells.

### 3.8. CD8^+^ T Cell-Dependent Induction of Apoptotic Cells Death in Murine Lupus Nephritis

Activation of CD8^+^ T cells by PTECs requires cell contact in the kidney. To assess localization of CD8^+^ T cells in murine immune-mediated GN, we stained for CD8 and the T-cell marker CD3 in serial kidney sections of healthy and diseased mice. The MRL-*lpr* mice were used as a model of systemic lupus erythematosus (SLE) to study localization of CD8^+^ T cells in lupus nephritis. At an age of 15 weeks, MRL-*lpr* mice showed elevated plasma autoantibody levels, an increased renal activity score [37], and proteinuria compared to MRL mice (Appendix A). In the naïve kidney, the number of CD8^+^ T cells was very low. We determined an elevated CD8^+^ T-cell number in the inflamed kidney, which predominantly localized in the tubulointerstitium. The number of CD8^+^ T cells within glomeruli was also increased although in lower numbers than in the tubulointerstitium of MRL-*lpr* mice (Figure 8A). We further assessed renal CD8^+^ T-cell localization in pristane-induced lupus nephritis. Here, pristane-treated WT mice developed autoantibodies and glomerular injury within nine months (Appendix A). Again, the majority of CD8^+^ T cells was localized in the tubulointerstitium of pristane-treated mice, and we detected only a low number within glomeruli (Figure 8B). Thus, in lupus nephritis, CD8^+^ T cells mainly accumulate in the tubulointerstitium, which enables close contact to PTECs.

We then analyzed the phenotype of renal CD8^+^ T cells in murine lupus nephritis. We detected elevated frequencies of CD8^+^ T cells expressing GzmB, CD107a, IFNγ, and IL-17A in MRL-*lpr* mice compared to MRL mice (Figure 8C). The same phenotype was observed in pristane-treated WT mice compared to age-matched naïve WT mice (Figure 8D), demonstrating a cytotoxic and an inflammatory phenotype of CD8^+^ T cells that infiltrate the kidney in lupus nephritis. As cytotoxic CD8^+^ T cells induce apoptosis in target cells, we performed a TUNEL staining to detect apoptotic cells in renal tissue. We determined an increased number of TUNEL^+^ cells in pristane-treated WT mice compared to healthy controls, which were mainly localized in the tubular system. Interestingly, lack of CD8^+^ T cells in *Cd8a*^-/-^ mice resulted in a reduced number of TUNEL^+^ tubular cells in lupus nephritis (Figure 8E). One mechanism by which GzmB induces apoptotic cell death is via activation of caspase-3 through proteolytic processing [55]. Therefore, we stained for cleaved caspase-3, the active form of this enzyme, in renal tissue. We found highly up-regulated expression of cleaved caspase-3 in pristane-treated WT mice compared to naïve mice. Particularly proximal tubuli showed expression of cleaved caspase-3, but we also detected an increased number of cleaved caspase-3^+^ cells within glomeruli. Induction of renal cleaved caspase-3 expression was strongly reduced in pristane-treated *Cd8a*^-/-^ mice in comparison to diseased WT mice (Figure 8F), further demonstrating the importance of CD8^+^ T cells for induction of apoptotic cell death in lupus nephritis. As MRL-*lpr* mice bear a mutation in the gene encoding Fas and are therefore not able to induce apoptotic cell death via the Fas/FasL pathway, which also targets caspase-3, the number of TUNEL^+^ cells and cleaved caspase-3^+^ cells were reduced in the kidneys of MRL-*lpr* mice compared to MRL mice (Appendix A), suggesting that apart from GzmB-induced cell death the Fas/FasL pathway also contributes to induction of apoptosis in lupus nephritis.

## 4. Discussion

Immune-mediated GN comprises a group of life-threatening diseases triggered by so far poorly defined mechanisms. Thus, identifying immunological pathways involved in disease pathology is of high clinical relevance. The outcome of an immune response depends on different factors such as the type of APC population, which initiates T-cell activation. While professional DCs promote CD8^+^ T cell-mediated immunity [12,15,16], non-professional LSECs favor induction of tolerance [22,23,24]. In this study, we described PTECs as a population of renal non-professional APCs that facilitate immunity by inducing cytotoxic and inflammatory CD8^+^ T cells via cross-presentation of soluble antigen in vitro. 

By performing expression analyses, we showed that PTECs expressed genes and proteins involved in receptor-mediated internalization of soluble antigen into endosomes, antigen translocation into cytosol, proteasomal antigen degradation, transport of antigen-derived peptides back into endosomes, peptide trimming and endosomal alkalization, altogether indicating that PTECs have the capacity for antigen cross-presentation via the endosome-to-cytosol pathway [25]. A comparison between mRNA and protein data revealed that mRNA abundances determined in the different cell populations only partially reflected the respective protein abundances. This is in line with studies demonstrating that mRNA levels can only explain to some extent variations in protein levels, which are differentially regulated in certain cell types. Ratios between mRNA and protein are mainly determined by protein translation and degradation. Protein levels also depend on protein function since less stable proteins allow rapid alterations in gene and subsequent protein responses to certain stimuli, e.g., inflammation [56,57]. Since we compared three different cell populations from which only BMDCs are classical APCs, while PTECs and LSECs also fulfil other organ-specific physiological functions, translation and degradation of the analyzed proteins may be differentially regulated in the cell populations leading to discrepancies in mRNA and protein abundances. 

Although PTECs expressed less MR than BMDCs and LSECs, we provided evidence that PTECs can use the MR for antigen uptake. By using fluorochrome-labelled OVA, we showed that PTECs internalized soluble antigen at 37 °C. Since OVA internalization was abrogated at 4 °C, this demonstrates an energy-dependent process and excludes unspecific binding of OVA on the cell surface. The MR has been implicated in soluble, mannosylated antigen internalization thereby promoting transport into endosomes for antigen cross-presentation [26,46]. We demonstrated that PTECs internalized soluble antigen via the MR—as blockage of the receptor by mannan decreased OVA uptake—whereas another sugar, which does not bind to the MR, did not affect antigen uptake. MR internalization occurs in clathrin-coated vesicles at the plasma membrane before transport to cytosolic endosomes. We showed that by preventing formation of clathrin-coated vesicles through CPZ [48], OVA uptake in PTECs was diminished. Based on these findings, we conclude that PTECs can internalize soluble antigen via MR-mediated endocytosis. Thus, PTECs exploit a well-described pathway of antigen uptake as DCs [47,58] and LSECs [59] have also been shown to internalize soluble antigen via the MR.

In the co-culture experiments using OVA protein as a soluble antigen, we showed an antigen-dependent activation and proliferation of CD8^+^ T cells in the presence of PTECs. Since in the controls without OVA or PTECs CD8^+^ T cells were not activated, this demonstrates the ability of PTECs to present a soluble antigen to CD8^+^ T cells thereby stimulating their activation. As a prerequisite, proteasomal degradation of internalized protein into antigenic peptides must be performed. We found that PTECs expressed the immunoproteasome-specific catalytic subunit LMP7, although its expression was diminished compared to BMDCs and LSECs. Pre-incubation of PTECs with the LMP7 inhibitor ONX 0914 resulted in a reduced activation and proliferation of CD8^+^ T cells. We showed that PTECs favor induction of immunity as PTEC-activated CD8^+^ T-cell expressed inflammatory cytokines, expressed the cytotoxic molecule GzmB and exerted a cytotoxic function by killing allogenic target cells. Both the expression of GzmB and the inflammatory cytokines were decreased after pre-treatment of PTECs with ONX 0914, suggesting that the immunoproteasome is involved in PTEC-mediated inflammatory CD8^+^ T-cell activation. However, our data showed that ONX 0914 not only bound to LMP7 but also to β5c thereby inhibiting the proteolytic activities of subunits that are present in the immunoproteasome or in the constitutive proteasome. This also explains the observed elevated protein abundance in ONX 0914-treated PTECs since inhibition of a catalytic subunit of the constitutive proteasome impairs regular protein degradation, leading to intracellular protein accumulation. Treatment of ONX 0914 further resulted in highly up-regulated expression of β1c in PTECs, without affecting its proteolytic activity. Since we also did not detect altered activity of β2c, another catalytic subunit of the constitutive proteasome, we conclude that PTECs can use LMP7 and β5c to generate antigenic peptides for cross-presentation to induce cytotoxic and inflammatory CD8^+^ T cells.

As shown for LSECs in the liver [22,23,24], non-professional APCs can induce another phenotype in activated CD8^+^ T cells than professional APCs, which influences the outcome of an immune response. We therefore compared the phenotype of PTEC-activated CD8^+^ T cells with those stimulated by splenic DCs or LSECs. After 2.5 days of co-culture, we found that all three APC populations induced expression of GzmB and inflammatory cytokines in CD8^+^ T cells. PTEC-induced CD8^+^ T-cell activation, proliferation, and IFNγ expression was reduced compared to DCs, illustrating the different capacities of professional and non-professional APCs for T-cell activation. In contrast, PTECs and LSECs did not differ in their potential to activate CD8^+^ T cells at this time point of analysis, but we determined a strongly reduced expression of IL-2 in LSEC-activated CD8^+^ T cells. Interestingly, there was a striking difference in the phenotype of activated CD8^+^ T cells after 5 days of co-culture. While PTEC- and DC-activated CD8^+^ T cells showed a sustained proliferation and expression of cytotoxic molecules and inflammatory cytokines, this was not the case for LSEC-stimulated CD8^+^ T cells. Here, CD8^+^ T cells no longer proliferated or expressed GzmB, and they also lost the capacity for degranulation, a prerequisite for the cytotoxic function. Moreover, they lacked expression of IL-2, and they showed strongly reduced production of IFNγ. These data are in line with previous studies, demonstrating functional inactivation of CD8^+^ T cells by LSECs [52,53,60,61]. Mechanistically, LSECs were shown to up-regulate expression of PD-L1 during co-culture with CD8^+^ T cells, thereby inducing a co-inhibitory signal in initially activated PD-1^+^ CD8^+^ T cells [52,53]. A low expression of co-stimulatory molecules and an impaired capacity to induce IL-2 expression in activated CD8^+^ T cells have also been identified as mechanisms of LSEC-induced CD8^+^ T-cell tolerance [53,60,61]. By comparing the expression of co-inhibitory and co-stimulatory molecules, we found that PTECs expressed less PD-L1 but much more CD80 and CD86 than LSECs, and in contrast to LSECs, they induced a strong and a sustained IL-2 expression in CD8^+^ T cells. These differences between PTECs and LSECs provide an explanation for the different outcome of CD8^+^ T-cell activation induced by non-professional APCs from the kidney and the liver. While LSECs limit CD8^+^ T-cell responses, thereby inducing tolerance against oral and tumor-derived antigens [22,23,24], PTECs support the induction of cytotoxic and inflammatory CD8^+^ T cells, which may contribute to the pathogenesis of immune-mediated GN.

In human lupus nephritis, CD8^+^ T cells have been shown to infiltrate the kidney [62,63], where accumulation of periglomerular CD8^+^ T cells correlated with disease severity [64], suggesting that they may contribute to disease pathology. In addition, renal CD8^+^ T cells expressing GzmB or another granzyme, GzmK, have been identified in lupus nephritis patients [65]. A pathogenic role of CD8^+^ T cells has also been described in different mouse models of immune-mediated GN [66]. However, how CD8^+^ T cells become activated in GN and the mechanisms by which they contribute to disease pathology is less clear. In murine lupus nephritis, the majority of kidney-infiltrating CD8^+^ T cells accumulated in the tubulointerstitium in close contact to proximal tubuli. Based on the in vitro studies, it may be conceivable that PTECs cross-present antigens in immune-mediated GN thereby promoting cytotoxic CD8^+^ T cells, which may induce apoptotic cell death in the tubular system, e.g., by release of GzmB, which thereby promotes tubulointerstitial nephritis. Our assumption that CD8^+^ T cells exert a cytotoxic function in immune-mediated GN is supported by the finding that the observed apoptotic cell death in the tubular system of mice with lupus nephritis was reduced in the absence of CD8^+^ T cells. Moreover, particularly PTECs showed an elevated expression of the active form of caspase-3 in lupus nephritis—an enzyme involved in the induction of apoptotic cell death—whose proteolytic activation can be induced by GzmB [55]. Interestingly, a lack of CD8^+^ T cells impaired activation of caspase-3 in PTECs, further indicating the key role of CD8^+^ T cells in the induction of apoptosis in murine lupus nephritis, which may mainly affect cross-presenting PTECs.

We also detected CD8^+^ T cells within glomeruli in lupus nephritis raising the question of whether they may contribute to glomerular damage in immune-mediated GN. Crescent formation is a hallmark of severe glomerular injury involving proliferation of cells in Bowman’s space as a result of damage in the capillary wall, glomerular basement membrane, and Bowman´s capsule, which lead to fibrin formation and induction of inflammatory immune responses [67]. The role of CD8^+^ T cells in these pathogenic processes is less clear. Two studies indicated that antigen-specific CD8^+^ T cells can infiltrate the glomeruli during autoimmune GN, if the Bowman´s capsule is destroyed, and kill intraglomerular cells such as podocytes, which present autoantigens [68,69]. Thus, cytotoxic CD8^+^ T cells may be responsible for the progression of glomerular damage in severe immune-mediated GN by killing antigen-presenting cells within glomeruli. Indeed, it has been shown that podocytes can function as renal APCs [70].

Cross-presentation of self-antigens has been implicated in the development of autoimmune diseases [18]. In one study, albumin was suggested as a self-antigen in renal disease with proteinuria, where breakdown of the glomerular filtration barrier results in access of albumin to kidney tissue. The authors demonstrated the generation of albumin-derived antigenic peptides through a concerted action of PTECs and DCs in vitro, resulting in DC-induced inflammatory CD8^+^ T-cell activation [71]. Whether PTECs are also capable of cross-presenting albumin-derived peptides has not been analyzed so far, but it may be possible that PTECs could thereby contribute to the development of autoimmune kidney disease.

In summary, this study revealed the capacity of PTECs for antigen cross-presentation thereby inducing cytotoxic and inflammatory CD8^+^ T cells, and it further identified CD8^+^ T cells to induce apoptosis in the tubular system in murine lupus nephritis. The predominant in vitro data open the possibility for hypotheses about the in vivo relevance of PTEC-mediated CD8^+^ T-cell activation for disease pathology of immune-mediated GN. However, it remains an important open question whether PTECs cross-present self-antigens in autoimmune kidney disease thereby becoming a target for cytotoxic CD8^+^ T cells.

## Figures and Tables

**Figure 1 cells-11-01510-f001:**
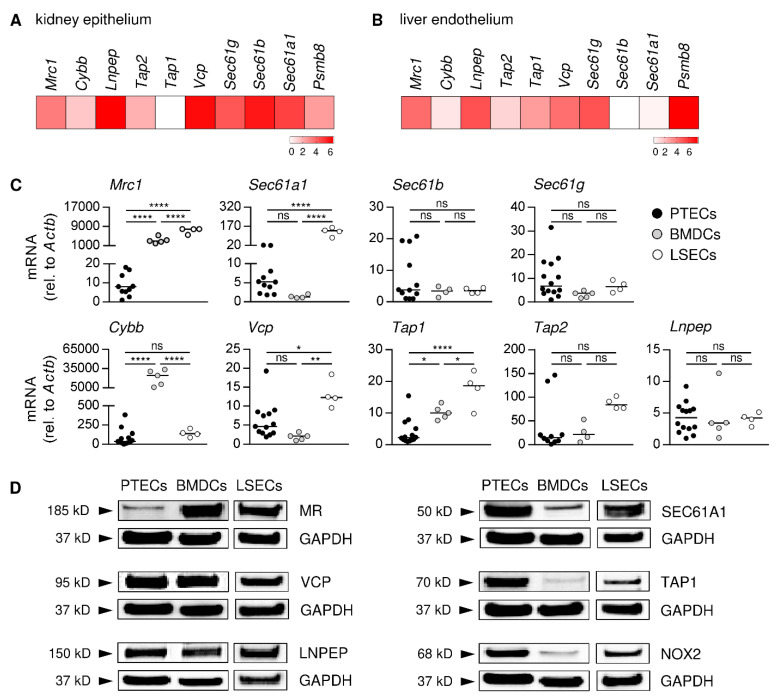
Gene and protein expression analysis of markers associated with antigen cross-presentation. RNA-seq data were used to create a gene set expression analysis of (**A**) renal epithelial cells and (**B**) liver endothelial cells. The Log_2_cpm data sets were depicted in heat maps. (**C**) Gene expression profiles of PTECs, DCs, and LSECs were assessed by quantitative RT-PCR and normalized to the reference gene β-actin (*Actb*). Medians of at least four experiments are shown. (**D**) PTEC, DC, and LSEC protein expression levels were determined by a WB analysis and depicted in relation to expression of the reference protein GAPDH. Blots are representative of two experiments. * *p*< 0.05; ** *p*< 0.01; **** *p*< 0.0001; ns: not significant.

**Figure 2 cells-11-01510-f002:**
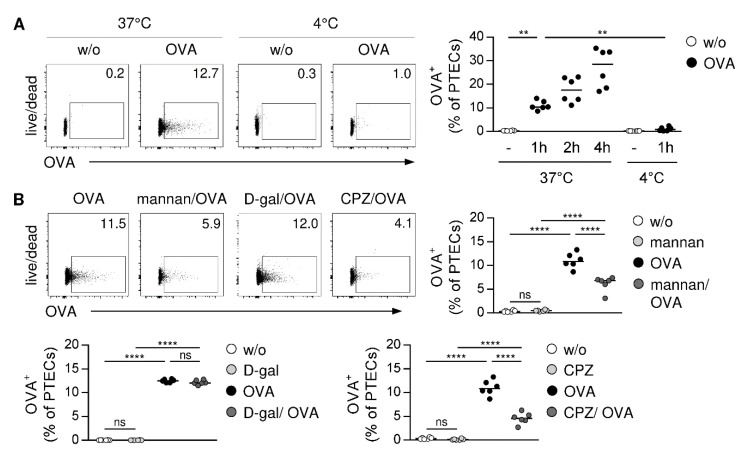
MR-mediated antigen uptake by PTECs. (**A**) PTECs were cultured in presence or absence of fluorochrome-labelled OVA at 37 °C or 4 °C over different periods of time and then analyzed by flow cytometry. (**B**) PTECs were pre-treated with mannan, D-galactose or CPZ before OVA incubation. Representative dot plots and medians of two experiments are shown. ** *p*< 0.01; **** *p*< 0.0001; ns: not significant; D-gal: D-galactose; CPZ: chlorpromazine; w/o: without.

**Figure 3 cells-11-01510-f003:**
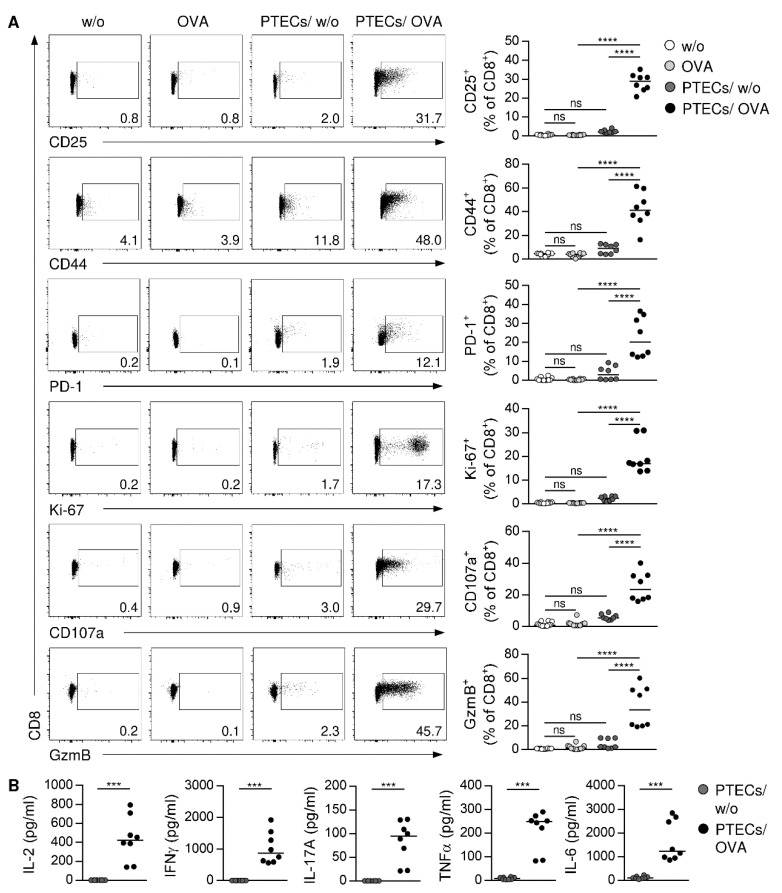
Antigen-dependent activation of CD8^+^ T cells by PTECs. OVA-specific CD8^+^ CD25^‒^ T cells were cultured with PTECs or alone in presence or absence of OVA for 2.5 days. (**A**) CD8^+^ T cells were stained for CD25, CD44, PD-1, Ki-67, CD107a, and GzmB, and analyzed by flow cytometry. (**B**) Cytokine levels were determined in culture supernatants. Representative dot plots pre-gated on CD8^+^ TCRβ^+^ cells and medians of 2–3 experiments are shown. *** *p* < 0.001; **** *p* < 0.0001; ns: not significant; nd: not detectable; w/o: without OVA.

**Figure 4 cells-11-01510-f004:**
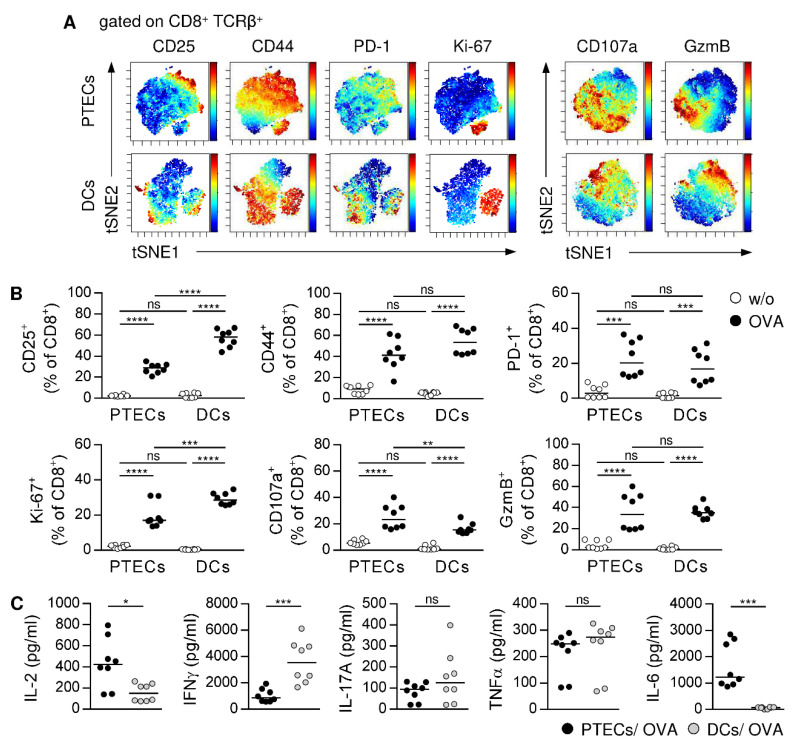
Phenotype analysis of CD8^+^ T cells activated by PTECs or splenic DCs. OVA-specific CD8^+^ CD25^‒^ T cells were cultured with PTECs or DCs in presence or absence of OVA for 2.5 days. (**A**) CD8^+^ T cells from co-cultures were stained for CD25, CD44, PD-1, and Ki-67 or CD107a and GzmB and analyzed by viSNE analysis. Single cells are represented by dots and expression levels are color coded from minimal (blue) to maximal (red). The viSNE analysis comprises 7955 PTEC-activated and 7829 DC-activated CD8^+^ T cells. (**B**) Phenotype of CD8^+^ T cells from co-cultures were analyzed by flow cytometry. (**C**) Cytokine levels were determined in co-culture supernatants. Representative tSNE plots and medians of two experiments are shown. * *p* < 0.05; ** *p* < 0.01; *** *p* < 0.001; **** *p* < 0.0001; ns: not significant; w/o: without OVA.

**Figure 5 cells-11-01510-f005:**
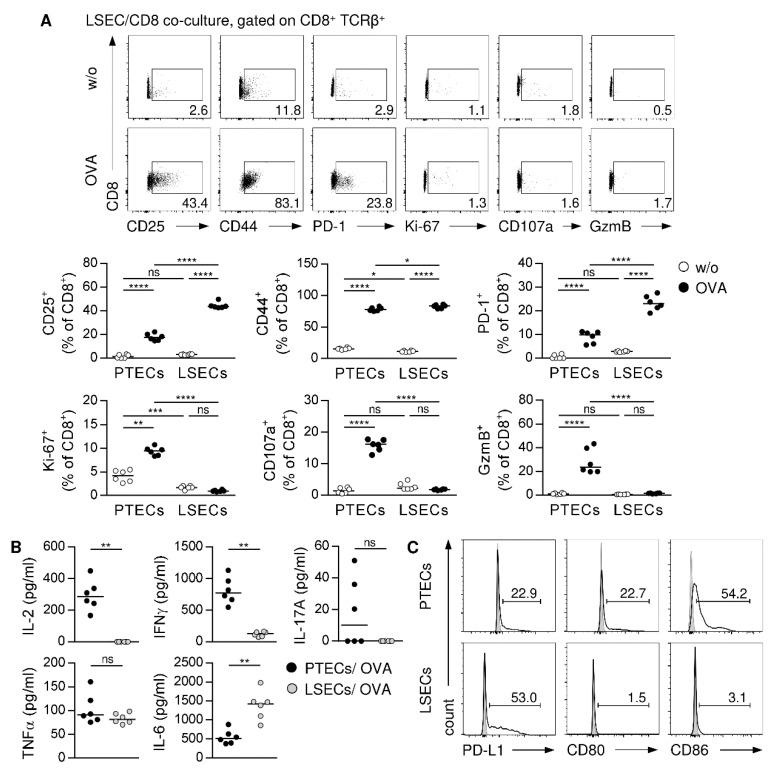
Phenotype analysis of CD8^+^ T cells activated by PTECs or LSECs. OVA-specific CD8^+^ CD25^-^ T cells were cultured with PTECs or LSECs in presence or absence of OVA for 5 days. (**A**) CD8^+^ T cells from co-cultures were stained for CD25, CD44, PD-1, Ki-67, GzmB, and CD107a and analyzed by flow cytometry. (**B**) Cytokine levels were determined in co-culture supernatants. Representative dot plots of LSEC/CD8 co-cultures and medians of two experiments are shown. (**C**) PTECs and LSECs were stained for PD-L1, CD80, and CD86 and analyzed by flow cytometry. Representative dot plots of two experiments are shown. * *p* < 0.05; ** *p* < 0.01; *** *p* < 0.001; **** *p* < 0.0001; ns: not significant; w/o: without OVA.

**Figure 6 cells-11-01510-f006:**
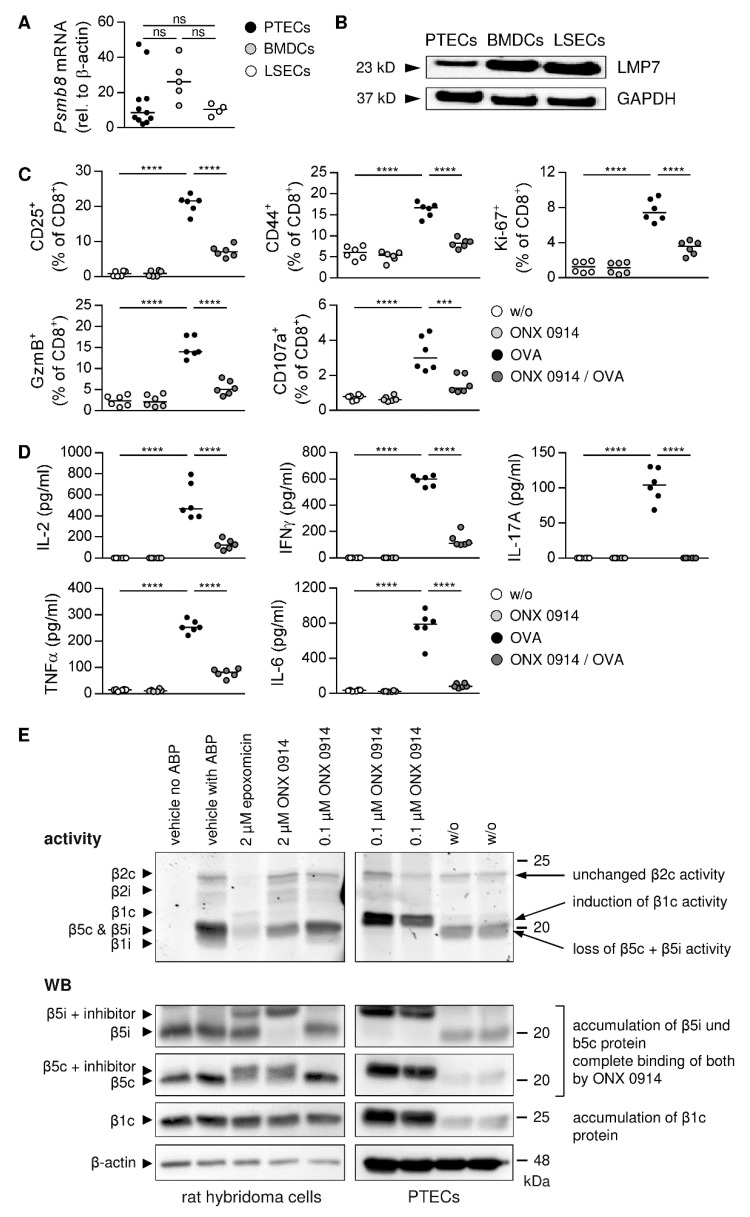
Proteasome-dependent CD8^+^ T-cell activation. (**A**) Expression of *Psmb8* was analyzed in PTECs, BMDCs, and LSECs by quantitative RT-PCR and normalized to the reference gene β-actin. Medians of at least four experiments are shown. (**B**) LMP7 protein expression was determined by WB analysis and depicted in relation to expression of the reference protein GAPDH. Blots are representative of two experiments. (**C**) PTECs were pre-incubated with ONX 0914 before co-culture with CD8^+^ T cells in presence and absence of OVA for 1.5 days. Phenotype of CD8^+^ T cells was analyzed by flow cytometry. (**D**) Cytokine levels were determined in culture supernatants. (**E**) PTECs were incubated with ONX 0914. Rat hybridoma cells were incubated with the proteasome inhibitor epoxomicin or ONX 0914. Activities of catalytic proteasome subunits were assessed by using ABPs. Protein expression of the different catalytic subunits was determined by WB analysis. Blots are representative of two experiments. Medians of two experiments are shown. *** *p* < 0.001; **** *p* < 0.0001; ns: not significant; w/o: without.

**Figure 7 cells-11-01510-f007:**
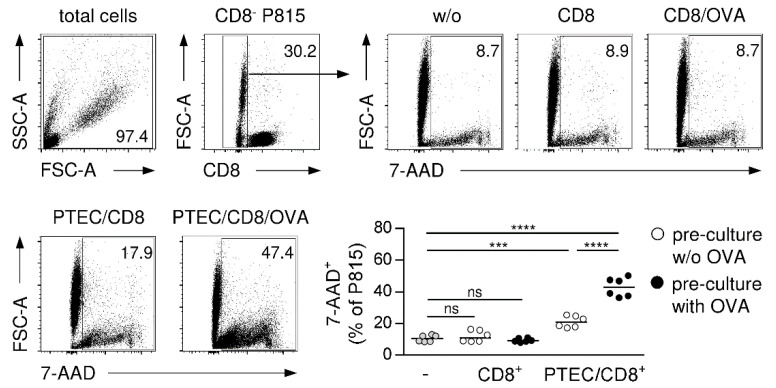
Cytotoxic function of PTEC-activated CD8^+^ T cells. CD8^+^ T cells pre-cultured with and without PTECs in presence or absence of OVA were incubated with allogenic P815 target cells for four hours. Cells were stained for 7-AAD and CD8 and analyzed by flow cytometry. Representative dot plots and medians of two experiments are shown. *** *p* < 0.001; **** *p* < 0.0001; ns: not significant; w/o: without; PTEC/CD8: CD8^+^ T cells pre-cultured with PTECs; CD8: CD8^+^ T cells pre-cultured alone.

**Figure 8 cells-11-01510-f008:**
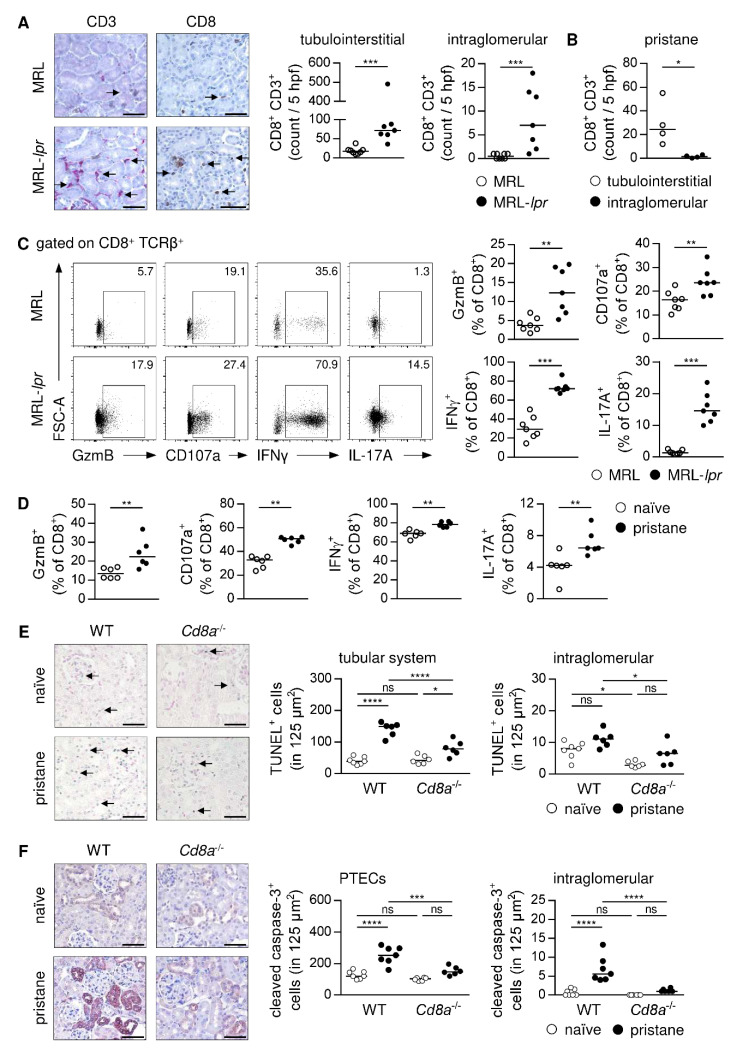
CD8^+^ T cell-dependent apoptosis in murine lupus nephritis. Serial kidney sections of (**A**) lupus-prone MRL-*lpr* mice and (**B**) pristane-treated WT mice were stained with anti-CD3 or anti-CD8 antibodies. Number of tubulointerstitial and intraglomerular CD8^+^ CD3^+^ T cells were counted in 5 hpf per section. Arrows mark CD8^+^ T cells. (**C**) Phenotype of renal CD8^+^ T cells was analyzed in MRL-*lpr* mice and (**D**) pristane-treated WT mice by flow cytometry. (**E**) Number and localization of renal apoptotic cells and (**F**) cleaved caspase-3^+^ cells were analyzed by TUNEL assay or cleaved caspase-3 staining in pristane-treated WT and *Cd8a*^-/-^ mice. Bars represent 50 µm. Representative dot plots and medians of one out of two experiments are shown. * *p* < 0.05; ** *p* < 0.01; *** *p* < 0.001; **** *p* < 0.0001, ns: not significant.

## Data Availability

All data generated or analyzed during this study are included in this published article and its Appendix A.

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
