# Peer review of "Antigen Cross-Presentation by Murine Proximal Tubular Epithelial Cells Induces Cytotoxic and Inflammatory CD8+ T Cells"

_cells, 2022, doi:10.3390/cells11091510_

Round 1
Reviewer 1 Report
Linke et al. reported that proximal tubular epithelial cells induce cytotoxic CD8+ cells by antigen cross-presentation, which might contribute to the pathophysiology of immune-mediated glomerulonephritis. Through this work, the antigen cross-presentation has an important role in immune-mediated glomerular disease by regulating the T cell activation and responses. This is well-designed experimental work. I have minor comments on this work.
- Immunohistochemistry data should improve their quality (Figure 8A, E, and F).
- Authors use the murine lupus nephritis model for immune-mediated glomerulonephritis. How about other kinds of glomerular disease such as the adriamycin-induced nephropathy model or renal ischemia-reperfusion injury model?
Author Response
Please find attached the point-by-point response for reviewer 1.

Reviewer 2 Report
The authors presented a well designed study looking at the role of proximal renal tubular epithelial cells in causing glomerulonephritis via induction of cytotoxic T cells via antigen cross presentation. Did the authors also look at the potential role of PTEC in induction of neutrophils which are also predominantly present in cases of glomerulonephritis in the glomerulus. Tubulointerstitial nephritis is potentially explained by this study but perhaps not the glomerular changes that are seen in immune mediated glomerulonephritis. The authors will need to modify the manuscript explaining this.
Author Response
Please find attached the point-by-point response for reviewer 2.

Reviewer 3 Report
Linke et al. investigated the capacity of PTECs for antigen cross-presentation; they concluded that PTECs have the capacity for antigen cross-presentation, thereby inducing cytotoxic CD8+ T cells in vitro, which might contribute to the pathology of immune-mediated glomerulonephritis. The study is novelty and designed well. The reviewer has some minor concerns:
- Figures 1, the bands of WB are overexposed. The results of the same protein should be displayed on the same gel and analyzed by densitometric analysis.
qPCR results of some genes were inconsistent with WB results, including VCP, LNPEP, TAP1, SEC61A1, and NOX2, why? How to explain it? This at least should be discussed.
- All results on Cytokine levels in culture supernatants should be normalized by the number of cells (total protein content in the cells).
- Figure 6E, the presented size of the band in the w/o group is small, why? It should be discussed.
- How about the general physiological data in Lupus-prone MRL-lpr and MRL control mice, including the urine excretion.
- It is suggested to add the figures to show the author's finding.
Author Response
Please find attached the point-by-point response for reviewer 3.

Round 2
Reviewer 2 Report
The authors have satisfactorily responded to the comments.